Expressional and functional interactions of two Apis cerana cerana olfactory receptors

Guo Lina 1 linaguo@126.com
Zhao Huiting 2
Jiang Yusuo 1
1 College of Animal Science and Veterinary Medicine, Shanxi Agricultural University , Taigu , China
2 College of Life Science, Shanxi Agricultural University , Taigu , China
Gould Gwyn
Electronic publication date: 2018 Jun 11
Publication date: 2018
Volume: 6
Electronic Location ID: e5005
Received 2018 Jan 8; Accepted 2018 May 29
Copyright: © 2018 Guo et al.
Copyright year: 2018
Copyright holder: Guo et al.
License: This is an open access article distributed under the terms of the Creative Commons Attribution License, which permits unrestricted use, distribution, reproduction and adaptation in any medium and for any purpose provided that it is properly attributed. For attribution, the original author(s), title, publication source (PeerJ) and either DOI or URL of the article must be cited.
License URL: https://creativecommons.org/licenses/by/4.0/

Keywords: Apis cerana cerana, Sf9 cells, Olfactory signaling, Ligand, Olfactory receptors

Funding: Shanxi Scholarship Council of China 2015-063 National Natural Science Foundation of China 31272513 This work was supported by the Shanxi Scholarship Council of China (Grant number 2015-063) and the National Natural Science Foundation of China (Grant number 31272513). The funders had no role in study design, data collection and analysis, decision to publish, or preparation of the manuscript.

==============================
Apis cerana cerana relies on its sensitive olfactory system to perform foraging activities in the surrounding environment. Olfactory receptors (ORs) are a primary requirement for odorant recognition and coding. However, the molecular recognition of volatile compounds with ORs in A. cerana cerana is still not clear. Hence, in the present study, we achieved transient transfection and cell surface expression of A. cerana cerana ORs (AcerOr1 and AcerOr2; AcerOr2 is orthologous to the co-receptor) in Spodoptera frugiperda (Sf9) cells. AcerOr2 narrowly responded to N-(4-ethylphenyl)-2-((4-ethyl-5-(3-pyridinyl)-4H-1,2,4-triazol-3-yl) thio) acetamide (VUAA1), whereas AcerOr1 was sensitive to eugenol, lauric acid, ocimene, 1-nonanol, linolenic acid, hexyl acetate, undecanoic acid, 1-octyl alcohol, and nerol. Of the compounds tested, AcerOr1 showed the highest sensitivity to these odorants with EC50 values of 10−7 and 10−8 M, and AcerOr2 recognized VUAA1 with higher sensitivity [EC50 = (6.621 ± 0.26) × 10−8]. These results indicate that AcerOr2 is an essential gene for olfactory signaling, and AcerOr1 is a broadly tuned receptor. We discovered ligands that were useful for probing receptor activity during odor stimulation and validated three of them by electroantennography. The response increased with concentration of the odorant. The present study provides insight into the mechanism of olfactory discrimination in A. cerana cerana.

Introduction

Insects recognize volatile substances in the surrounding environment using olfactory receptors (ORs). These receptors bind different odorant molecules to produce a signaling cascade that opens ion channels and sends electronic messages from the olfactory sensory neurons (OSNs) to the brain. There are two models of odorant signaling pathways in insects. One study showed that odorant receptors (composed of Orx and Orco) form heteromeric ligand-gated ion channels that that are directly gated by odorants (Sato et al., 2008). Another mechanism suggests that after odorant binding, the activity is transferred to the Orco subunit through direct (fast and short) ionotropic or indirect (slow and long) G-protein-coupled receptors (GPCR)-based metabotropic pathways. In the direct pathway, odorant binding directly opens a channel formed by the Orco subunit, generating a fast and short ionotropic depolarization response. By contrast, in the indirect pathway, Orx activates the synthesis of cAMP through a G protein (Gs) and an adenylyl cyclase, and this in turn activates Orco, leading to cyclic cAMP production. Upon binding of cAMP to Orco, the channel opens and generates a slow and prolonged depolarizing response (Wicher et al., 2008).

Compared with mammalian GPCR, the seven transmembrane domains (TMDs) in insect ORs are inverted, where the C-terminus is extracellular and the N-terminus is intracellular, and there is no homology between the two proteins (Benton et al., 2006; Missbach et al., 2014). In vivo, OR proteins are expressed primarily in the OSNs of the antennae (de Bruyne, Foster & Carlson, 2001; Vosshall et al., 1999), and olfaction requires co-expression of a highly conserved common olfactory receptor subunit (Orco) and variable odorant-binding subunits (ORx) in each OSN (Benton et al., 2006; Carey et al., 2010; Larsson et al., 2004; Neuhaus et al., 2005; Vosshall & Hansson, 2011; Wang et al., 2010). Orco does not bind odorants but is necessary for localization, stabilization, and correct protein folding of conventional ORs to dendritic membranes, and it is highly conserved in different species. Conversely, the function of ORs has generally been considered to be interaction with a select range of odorants (Carey et al., 2010; Claudianos et al., 2014; Wang et al., 2010), and these receptors are much more divergent both within and across species (Hill et al., 2002; Jordan et al., 2009; Krieger et al., 2003; Malpel et al., 2008; Melo et al., 2004; Shen et al., 2011; Xia & Zwiebel, 2006).

With the widespread emergence of genome sequencing, many ORs have been identified; however, the identification of their ligands still lags behind, causing most of the ORs to become orphan receptors. For example, there are 119 ORs in Apis cerana (Park et al., 2015), and more than 177 in A. mellifera (Robertson & Wanner, 2006; Wanner et al., 2007) based on honeybee genome analysis, but only a few studies have shown that the honeybee OR Or151 is expressed at higher levels in worker bees and binds to the floral odorant linalool (Reinhard & Claudianos, 2012). AmOr11 is specifically binds to the queen pheromone 9-oxo-decenoic acid (9-ODA) (Wanner et al., 2007), indicating that the OR binding odorant in honeybees associated with the scent detection environment.

AcerOr2, which is orthologous to the co-receptor, was expressed in all development stages of worker antennae in A. cerana (Zhao et al., 2013), and a similar expression profile was found for the A. mellifera Orco gene AmelOr2 (Krieger et al., 2003). AcerOr1 is expressed in both worker and drone antennae at different developmental stages (Zhao et al., 2014). These observations suggest that AcerOr2 and AcerOr1 are involved in sensory processes and may be caste- or task-dependent.

AcerOr1 and AcerOr2 were chosen for functional analysis (GenBank accession numbers: AcerOr1 (JN544932), AcerOr2 (JN792581)) using a heterologous expression system in Spodoptera frugiperda (Sf9) cells. When odorants interact with ORs in the cell surface, signal transduction occurs. We screened a panel of 33 odorants and determined the molecular receptive range of AcerOr1 and AcerOr2. We then used an electroantennography (EAG) assay to determine how physiological responses of odorants in honey bees affect their behavioral responses. Overall, the assessment of the functional properties of AcerOrs improves the current understanding of the mechanism of olfactory regulation in A. cerana cerana.

Materials and Methods

Odors

All odorants used in the present study were purchased from (Sigma-Aldrich, St. Louis, MO, USA) and were of the purest grade (>95% pure). Stock solutions (100 mM) of the odorants were prepared using dimethyl sulfoxide (DMSO) and stored at −20 °C. For each assay, odorant solutions were freshly diluted from the stock solution to the desired concentration in DMSO. Fluo-4-(acetoxymethyl) ester (Fluo-4 AM) (excitation at 494 nm, emission at 516 nm), obtained from (Beyotime, Shanghai, China) as a lyophilized powder, was diluted to 1 mM using DMSO and stored at −20 °C. The composition of the calcium assay buffer was as follows: 21 mM KCl, 12 mM NaCl, 18 mM MgCl2, 3 mM CaCl2, 170 mM D-glucose, 1 mM probenecid (Sigma-Aldrich, St. Louis, MO, USA), and 10 mM piperazine-1,4-bisethanesulfonic acid. The pH of the buffer was adjusted to 7.2, and the buffer was filter-sterilized (using a 0.22 µm filter) prior to use.

Vector construction

The pIB-AcerOr1/pIB-AcerOr2 plasmid constructs containing intact open reading frames, which were amplified with specific primers containing BamHI and EcoRI (NEB, Beverly, MA, USA) sites for the A. cerana cerana ORs AcerOr1 and AcerOr2 and cloned in the multiple cloning site of the pIB/V5-His vector (Invitrogen, Carlsbad, CA, USA), were used to generate the final transformation plasmids by restriction digestion with BamHI and EcoRI (NEB, Beverly, MA, USA). The specific primer sequences were as follows: AcerOr1 F: 5′-CGCGGATCCATGGAAAATACCACGAATTATCGTA-3′, AcerOr1 R: 5′-CCGGAATTCTACCGTCATTGCACGCAGAA-3′, AcerOr2 F: 5′-CGCGGATCCATGATGAAGTTCAAGCAACAGGG-3′, AcerOr2 R: 5′-CCGGAATTCCTTCAGTTGCACCAACACCA-3′.

Cell culture and transfection of Sf9 cells

Spodoptera frugiperda Sf9 cells were purchased from the Chinese Academy of Sciences Cell Bank (Beyotime, Shanghai, China) were maintained as an adherent culture in Sf-900 III serum-free medium (SFM; Gibco, Invitrogen, Carlsbad, CA, USA) supplemented with 10% fetal bovine serum (Sijiqing, Hangzhou, China) and 100 μg mL−1 penicillin-streptomycin at a constant temperature of 28 °C in a humidified incubator (Thermo Scientific, Cornelius, OR, USA) in the absence of CO2 in T-25 tissue culture flasks (Corning Inc., Corning, NY, USA). Sf9 cells were grown to approximately 80–90% confluence as observed under a light microscope. Cells were dislodged from the flask by washing with the media contained in the flask. In total, 1 × 106 Sf9 cells were suspended in 2 mL of Sf-900 III SFM in each well of a Nunclone six-well tissue culture plate (Corning Inc., Corning, NY, USA). Confluent cells (80–90%) were transiently transfected with 2.0 μg pIB-AcerOr1/pIB-AcerOr2 using 8 μL of Cellfectin II reagent (Invitrogen, Carlsbad, CA, USA) in six-well plates according to the manufacturer’s instructions. The medium containing plasmid DNA and Cellfectin II was removed after incubation of the cells with a DNA/Cellfectin II mix for 3–5 h. The cells were washed twice with fresh Sf-900 III SFM and overlaid with 2 mL of fresh SFM. G418 was used to select stably transfected cell lines, and the optimum G418 concentration for screening the stable cell lines was 400 μg mL−1. After incubation for 48 h, assays were performed.

Western blot and immunofluorescence analysis

To acquire polyclonal antigens (pAb_AcerOr1 and pAb_AcerOr2) of high titer and specificity against OR AcerOr1 and AcerOr2, the primary amino acid sequence and secondary protein structure information for AcerOr1 and AcerOr2 in NCBI GenBank was used, and one sequence-specific polypeptide was obtained using the bioinformatics software BLASTn, BLASTx, ExPASy, DNASTAR, and ANTHEPROT. These programs were used to analyze the amino acid sequence features of AcerOr1 and AcerOr2 proteins, including their hydrophilicity, flexibility, surface probability and antigenicity, and their secondary structures, and to find the antigenic peptides AcerOr1 ENTTNYRNIHYKSD (14 aa) and AcerOr2 NARYHQIAVK (10 aa). An AcerOrs antibody made by AbMax (AbMax Biotechnology Co., Ltd., Beijing, China) was used for western blot analysis and immunostaining to confirm the expression of AcerOr1 and AcerOr2.

Goat anti-Rabbit IgG, Alexa Fluor 488/594, and 4′,6-diamidino-2-phenylindole (DAPI) (Beyotime, Shanghai, China) were used to stain AcerOr1/AcerOr2 in transfected Sf9 cells grown on poly-l-lysine-coated coverslips placed in six-well plates. Thereafter, the medium was removed from the wells, and the cells were washed with phosphate-buffered saline and fixed with 4% (v/v) paraformaldehyde for 30 min at 25 °C. They were then treated with 5% bovine serum albumin for 1 h at 25 °C. Subsequently, the sections were incubated with rabbit anti-AcerOr (1:2,000 dilution) polyclonal antibody overnight at 4 °C. The secondary antibody (goat anti-Rabbit Alexa Fluor 488 or Alexa Fluor 594; 1:10,000) was applied for 2 h at 37 °C. Next, cells were incubated with 1 mL of DAPI (1:10,000). The coverslips with the stained cells were removed for analysis using immunofluorescence microscopy. Images were analyzed using the ImageJ software (National Institute of Health, Bethesda, MD, USA).

For western blotting, protein was extracted from cells expressing plasmids and transfected with either AcerOr1 or AcerOr2 or co-transfected with AcerOr1 and AcerOr2 using a cell lysis buffer (1% NP-40, 0.5% sodium deoxycholate, 0.1% SDS). The extracted proteins (100 μg per sample) were separated by 12% SDS-PAGE and transferred onto a nitrocellulose filter membrane (Boster, Wuhan, China). Membranes were blocked for 1.5 h at 25 °C in 5% skim milk (Boster, Wuhan, China), washed with Tris-buffered saline containing Tween-20 (TBST, pH 8.0), and incubated overnight at 4 °C with rabbit polyclonal anti-AcerOr, mouse anti-His-tagged (1:1,000 (v/v)) (BioWorld, Minneapolis, MN, USA) and mouse anti-β-actin (1:500 (v/v)) (Boster, Wuhan, China) antibodies. Thereafter, membranes were washed with TBST and incubated with the secondary antibodies, namely horseradish peroxidase-conjugated donkey anti-rabbit IgG (1:5,000 (v/v)) (Boster, Wuhan, China) and goat anti-mouse (1:2,000 (v/v)) IgG (Boster, Wuhan, China), respectively, for 2 h at 25 °C. Finally, membranes were washed three times with TBST. Bands were detected using Super ECL Plus detection reagent (Boster, Wuhan, China), and the western blot signal was analyzed using Image Lab (Bio-Rad Laboratories, Hercules, CA, USA) and Image J 1.49.

Ca2+ imaging

To identify candidate ligands for selected ORs, we tested 32 compounds (most of which were volatile compounds from host plants, including aldehydes, alcohols, monoterpenes, benzoates, and sesquiterpenes) at a final concentration of 10−6 M by Ca2+ imaging in an in vitro cell expression system. Thereafter, we determined the concentration-response curves for ten compounds (selected from the 32 compounds) and calculated their half-maximal effective concentration (EC50) values. After the cells were transfected for 48 h, the medium was removed and the cells were washed three times with Hank’s balanced salt solution (without Ca2+). The cells were subsequently cultured at 37 °C in the dark for 30 min in the presence of 2 μmol L−1 Fluo-4-AM (Beyotime, Shanghai, China) and were stimulated by the chemical odorants. Each test chemical ligand was applied to Fluo-4-loaded Sf9 cells expressing AcerOr1 at a final concentration of 10−6 M, and the increase in fluorescence caused by the substrate was measured and expressed as a fraction of the fluorescence elicited by the calcium ionophore ionomycin. The Ca2+-free solution used was Dulbecco’s phosphate buffered saline supplemented with 0.4 mM ethylene glycol tetra-acetic acid (EGTA). Fluorescence was measured using excitation and emission wavelengths of 494 and 516 nm, respectively, and the results were recorded by a Synergy H1 microplate reader (BioTek, Winooski, VT, USA). The formula used for calculating the free intracellular Ca2+ concentration was as follows: [Ca2+]i=Kd(F−FminFmax−F)

where Fmin and Fmax are the minimum fluorescence values under Ca2+-saturating conditions in the presence of 5 μM A23187 (a Ca2+-ionophore) and the maximum fluorescence values under zero-Ca2+ conditions when 4 mM EGTA was used in combination with 5 μM A23187, respectively. Kd is the dissociation constant of Fluo-4/Ca2+ (360 nM).

EC50 values were determined using GraphPad Prism 6.0 (GraphPad; San Diego, CA, USA). The formula used for calculating the concentration-response using a 4-parameter logistic model was as follows: Y=Bottom+(Top−Bottom)1+10(LogEC50−X)*HillSlope

where Y and X are the response showing a sigmoid shape and the logarithm of the concentration. EC50 is the concentration of odorant yielding 50% of its maximal effects, and HillSlope is the slope parameter. Residual standard deviation, the coefficient of determination (Xu et al., 2012), and 95% confidence intervals were calculated to verify that the fitted curve was correct.

Electroantennography

The level of OR expression in antenna olfactory neurons can be indirectly measured by recording the EAG responses of the isolated antennae to the corresponding odorant ligand. We compared data previously obtained by Ca2+ imaging in vitro to determine whether these odorants can cause electrophysiological responses in the antennae with physiological activity. Based on the results of the Ca2+ assay, three volatile compounds (VUAA1, eugenol, and linolenic acid) were used to record antennal responses. The compounds were dissolved and diluted in liquid paraffin to final concentrations of 0.1, 1, 10, 100, and 500 μg μL−1. Pure liquid paraffin wax was used as a blank, and results were calculated relative to the blank. The antennae were carefully cut at the base and placed into EAG electrode probes (Syntech, Hilversum, The Netherlands) with a drop of Spectra 360 electrode gel (Parker Lab, Inc., Fairfield, NJ, USA). Filter paper strips (5 × 50 mm) were loaded with 20 μL of the different test solutions and inserted into glass Pasteur pipettes and served as sources of stimuli. Humidified airflow was delivered at a constant rate of 700 mL min−1 by an air stimulus controller CS-55 (Syntech, Kirchzarten, Germany). Odor stimuli were administered three times at 2 mL s−1 for 0.5 s at 30 s intervals. EAG recordings of antennal responses to each stimulus were documented as voltage waveforms using an IDAC-4 computer-operated amplifier controller (Syntech, Kirchzarten, Germany), and the data were analyzed with the EAGPro software (Syntech, Kirchzarten,Germany). A newly prepared antenna was used for each recording. A dose-response curve was plotted using the EAG recordings (in mV) for each concentration.

Statistical analysis

The TMDs were predicted using TTHMM server v.2.0 and HMMTOP. Data were analyzed with SPSS v17.0 (SPSS Inc., Chicago, IL, USA) and expressed as the mean ± standard error (SEM). t-tests, ANOVAs, and Duncan’s multiple range tests were used to determine whether differences in the mRNA and protein levels or the EAG responses of antennae were significantly different among treatments. In all cases, statistical significance was tested at the 0.05 level.

Results

Membrane topology analysis of the AcerOrs

We first analyzed the primary amino acid sequence of the A. cerana cerana AcerOrs protein. We identified that AcerOr1 did not have the putative CaM binding site and gate sequences, and we thus selected the sequence ENTTNYRNIHYKSD (14 aa) in the N-terminal hydrophobic area as the multiple antigen peptide of AcerOr1 (Fig. 1A). A candidate calmodulin (CaM)-binding amino acid motif 328SAIKYWVER336 within the second intracellular loop (ICL2) of the AcerOr2 protein and, the sequence NARYHQIAVK (10 aa) in the ICL1 domain served as the multiple antigen peptide of AcerOr2 (Fig. 1B).

Figure 1 Predicted membrane topology of AcerOrs.

(A) Transmembrane regions of AcerOr1 and (B) AcerOr2 were predicted using TTHMM and HMMTOP. Gray circles indicate the amino acid sequences of the two receptors and highlight the regions that were targeted for the generation of the antibody; red circles indicated the CaM binding site.

Heterologous expression and localization of AcerOr1 and AcerOr2 in Sf9 cells

To confirm that ORs were successfully transfected in the Sf9 cells, the cells was stained with anti-AcerOr1 or anti-AcerOr2 followed by goat anti-rabbit Alexa Fluor 488 or 594. The results revealed that both AcerOr1 and AcerOr2 were expressed and localized to the plasma membrane of Sf9 cells (Fig. 2A), as pIB/V5-His-transfected Sf9 cells showed no immunostaining. Western blotting of Sf9 cell extracts using an anti-AcerOr1 and anti-AcerOr2 antibody revealed a specific band of approximately 50 and 56 kDa in Sf9 cells transfected with pIB/V5-AcerOr1 and pIB/V5-AcerOr2, but no specific bands in the Sf9 cells (negative control) or pIB/V5-His-transfected Sf9 cells (Fig. 2B). These results confirmed successful construction of the recombinant plasmids and expression of the corresponding OR in the Sf9 cells after in vitro transfection.

Figure 2 Western blot analysis and immunostaining of AcerOr1 and AcerOr2.

(A) Cells transfected with pIB/V5-His as a control DNA construct showed no staining, whereas Sf9 cells transiently transfected with pIB-AcerOr1 and immunostained with pAb-AcerOr1 showed strong staining with Alexa 488 (green) for cells expressing AcerOr1, and Sf9 cells transiently transfected with pIB-AcerOr2 and immunostained with pAb-AcerOr2 showed strong staining with Alexa 594 (red) staining for cells expressing AcerOr2. Nuclei were stained with DAPI. Scale bar = 50 μm. (B) Recombinant His-tagged AcerOr1 and AcerOr2 expression levels were determined by western blotting in non-transfected Sf9 cells and cells transfected with AcerOr1 and AcerOr2. The molecular weights of AcerOr1, AcerOr2 and β-actin are 50, 56, and 43 kDa, respectively. All experiments were repeated three times; the images are from the same sample re-runs.

Screening of specific ligands of AcerOrs

To test the potential functional activity of AcerOr during the olfaction process, odorant ligand binding is essential. The transfected cells were tested against a panel of 32 odorants (Fig. 3). The control cells transfected with the pIB/V5 empty vector were indifferent to all odorants tested (Fig. 3A). Nine of the 32 compounds, including eugenol, lauric acid, ocimene, 1-nonanol, linolenic acid, hexyl acetate, undecanoic acid, 1-octyl alcohol, and nerol, elicited responses from AcerOr1-expressing cells when administered at the high concentration of 10−6 M. Each of these odor compounds activated AcerOr1 to differing levels as reflected by differing fluorescence intensities (Fig. 3B). AcerOr2 expressed alone did not show responses to any floral odorants except its agonist VUAA1, which yielded the greatest increase in intracellular Ca2+, whereas those expressing AcerOr1 alone were sensitive to all nine volatile compounds mentioned above and VUAA1 (Fig. 3).

Figure 3 Response profile of Fluo-4-loaded Sf9 cells transfected with pIB-AcerOr1 and pIB-AcerOr2 to various odorants (10−6 M) using calcium imaging.

(A) Cells transfected with the pIB/V5 empty vector (as a control), (B) pIB-AcerOr1, or (C) pIB-AcerOr2 were stimulated by different odorants as indicated. For each Sf9 cell, [Ca2+]i was calculated as the maximum increase in [Ca2+]i obtained for an odorant minus the [Ca2+]i in the resting state. Bars indicate the mean ± SEM. (n = 3 for each odorant tested).

To examine the sensitivity of AcerOr for the above-described selected ligands, calcium assays were conducted for a range of ligand concentrations to support the construction of concentration-response curves with calculated half-maximal effective concentration (EC50) values. Ligand concentrations were reduced by 10-fold starting at 10−5 M. We then determined the activity of the ten compounds in more detail by constructing concentration-response curves for activation of AcerOr1 and AcerOr2 alone (Fig. 4). For AcerOr1 activation, the EC50 values ranged from 10−7 to 10−8 M for VUAA1, eugenol, lauric acid, ocimene, 1-nonanol, linolenic acid, hexyl acetate, undecanoic acid, 1-octyl alcohol, and nerol; for AcerOr2 activation, the EC50 value of VUAA1 was (6.621 ± 0.26) × 10−8 M (Fig. 4; Table 1).

Figure 4 Concentration response analysis for cells expressing AcerOr1 and AcerOr2.

(A) Concentration-response curve of AcerOr1 for ten compounds (VUAA1, undecanoic acid, eugenol, ocimene, nerol, 1-octyl alcohol, 1-nonanol, hexyl acetate, linolenic acid, lauricacid) and that of (B) AcerOr2 with VUAA1 for ten compounds (VUAA1, undecanoic acid, eugenol, ocimene, nerol, 1-octyl alcohol, 1-nonanol, hexyl acetate, linolenic acid, lauricacid) based on Ca2+-imaging assays. the value of [Ca2+]i (the concentration of Ca2+) was calculated. This value represented the maximum increase in [Ca2+]i obtained for an odorant minus the [Ca2+]i in the resting state. Bars indicate the standard deviation based on three biological replicates. The data were fitted by the 4-parameter logistic model from GraphPad Prism 6.0. Data points represent means ± SEM. Responses have been normalized after the addition of odorant ligand compared with before ligand addition. EC50 values for AcerOr1 and AcerOr2 can be found in Table 1.

Table 1 EC50 of different odorants for cells expressing AcerOr1 or AcerOr2.

	AcerOr1	AcerOr2	
	Emax	HillSlope	EC50 (M)	Emax	HillSlope	EC50 (M)	
VUAA1	4.64 ± 0.10	1.72 ± 1.48	(1.513 ± 0.122) × 10−7	241.67 ± 0.74	0.56 ± 0.08	(6.621 ± 0.26) × 10−8	
Eugenol	88.53 ± 0.52	0.74 ± 0.06	(1.02 ± 0.15) × 10−7	NR	NR	NR	
Lauric acid	138.79 ± 0.5	0.49 ± 0.06	(4.811 ± 2.05) × 10−8	NR	NR	NR	
Ocimene	144.27 ± 0.59	0.50 ± 0.09	(5.322 ± 0.39) × 10−8	NR	NR	NR	
1-Nonanol	69.86 ± 0.44	0.80 ± 0.06	(6.327 ± 1.25) × 10−7	NR	NR	NR	
Linolenic acid	85.74 ± 0.83	0.87 ± 0.03	(6.175 ± 0.22) × 10−7	NR	NR	NR	
Hexyl acetate	115.94 ± 1.5	0.73 ± 0.05	(1.008 ± 0.02) × 10−7	NR	NR	NR	
Undecanoic acid	116.69 ± 1.15	0.73 ± 0.05	(7.357 ± 0.04) × 10−8	NR	NR	NR	
1-Octyl alcohol	106.77 ± 2.01	0.63 ± 0.04	(8.125 ± 0.12) × 10−8	NR	NR	NR	
Nerol	184.92 ± 0.55	0.52 ± 0.06	(4.34 ± 0.06) × 10−8	NR	NR	NR	
Notes:

EC50 values were calculated using GraphPad Prism 6 software to plot and fit data to curves using a 4-parameter logistic model with the following equations: Y=Bottom+(Top−Bottom)/1+10(LogEC50−X)*HillSlope.

NR, no detectable response.

Electrophysiological response of Apis cerana cerana antennae

The selected odorants were tested for their the ability to elicit EAG responses in the antennae of A. cerana cerana, and three floral volatiles (VUAA1, eugenol, and linolenic acid) caused irritation and elicited EAG responses (Fig. 5). All three compounds showed a dosage-dependent increase in EAG response, and the highest effect was observed at a high concentration of 500 μg μL−1 of compound (18.71 ± 1.45, 9.92 ± 0.58, 10 ± 0.54 mV, respectively.) compared with 0.1, 1, 10, and 100 μg μL−1, (P < 0.05; Fig. 5). These results were consistent with those for Ca2+ imaging.

Figure 5 Relative electroantennogram (EAG) responses of Apis cerana cerana to three volatile odorants at different doses.

Experiments were repeated three times, and EAG recordings from ten antennae per group were obtained. Bars represent the means ± SEM based on a one-way ANOVA with the Duncan’s test (n = 3).

Discussion

The activity of AcerOrs is regulated by their ligands, and the discovery of ligand specificity and the physiological significance of AcerOrs is of great significance in understanding how the olfactory genome encodes the effect of odors on foraging behavior in the honey bee olfaction system.

Several signals regulate the opening and closing of ion channels (e.g., protein–protein interactions, ligand binding, membrane electrochemical gradients, and post-translational modifications); these signals are thought to cause conformational changes of the ion channels to expand or contract the gate (daCosta & Baenziger, 2013; Zhou & McCammon, 2010). Based on membrane topology analysis of the AcerOrc2 sequences, we showed that the CaM-binding domain residue in ICL2 was found to be highly conserved. This domain has been studied in Drosophila melanogaster 336SAIKYWVER344 (Mukunda et al., 2014) and in Aedes albopictus, 329SAIKYWVER337 (Liu et al., 2016), and this sequence conservation indicates that CaM activity in ICL2 plays a crucial functional role in the formation of functional ion channels by AcerOr2. In addition, the Tyr residue Y469 in AcerOr2 TM7 is conserved in Bombyx mori (Y464) (Nakagawa et al., 2012) and Leucozona lucorum (Y463) (Zhou et al., 2014) and has been confirmed to be critical for ion function, whereas conventional AcerOr1 does not have this sequence. Both the AcerOr1 and AcerOr2 proteins belong to TM7 domains and have an extracellular C-terminus, which consist with the membrane topology predictions of other insect OR and contrasts with the mammalian GPCR (Benton et al., 2006; Liu et al., 2016; Missbach et al., 2014).

The presence of AcerOr1 and AcerOr2 at the plasma membrane and protein levels measured upon heterologous expression in the Sf9 cells (detected by immunofluorescence and western blot analysis, respectively) suggested that AcerOrs were successfully transfected in the Sf9 cells, indicating that a functional assay is feasible. The functional assays and imaging involving Sf9 cells were performed as described previously (Kiely et al., 2007). Sf9 cells originate from the ovarian tissue of moth species, are known to express SfruOrco, and an ortholog of the co-receptor (Or83b) has been identified in Sf9 cells (Smart et al., 2008).

Orco has the function of supporting the dendritic localization of OrX proteins (Larsson et al., 2004), and it participates in the formation of OR ion channel pores (Nichols, Chen & Luetje, 2011; Pask et al., 2011) and forms a homologous ion channel without OrX (Wicher et al., 2008). Our results suggest that endogenous Orco supports the dendritic localization of AcerOr1 and forms a homologous ion channel with the ortholog of the co-receptor AcerOr2.

Critical parameters include the number and selection of odorants tested in the functional assay and how this panel covers the range of chemicals recognized by the receptor. Using Sf9 cells, we found that the cells transfected with the pIB/V5 empty vector were indifferent to all odorants tested, and that AcerOr2 alone does not confer floral odorant sensitivity, whereas AcerOr1 was functionally characterized as a receptor for general floral scents.

The results indicate that AcerOr1 accepts a broad range of odorants, even structurally unrelated odorants such as alcohol, benzoate, ester, and aliphatic acid. A previous study also showed that certain ORs bind a wide range of structurally unrelated odorants; for example, locust LmigOR3 was broadly tuned to esters, ketones and heterocyclic compounds (You et al., 2016). In contrast, some odorant receptors are narrowly tuned to odorants; for example, CquiOR10 is narrowly tuned to skatole in mosquitos (Hughes et al., 2010), AmOr11 is highly specific for the queen pheromone 9-ODA (Wanner et al., 2007), and OscaOR4 has high specificity for (E)-11-tetradecenyl acetate (Miura et al., 2010).

In our study, when we did not transfect AcerOr2, endogenous Orco in the Sf9 cells was unable to respond to the co-receptor agonist VUAA1. However, when AcerOr1 was expressed in the presence of endogenous Orco, non-specific cation channels were activated through some direct or indirect mechanisms in response to a range of floral odorants including eugenol, lauric acid, ocimene, 1-nonanol, linolenic acid, hexyl acetate, undecanoic acid, 1-octyl alcohol, and nerol. This may indicate that the endogenous Orco in Sf9 cells is insensitive to the Orco agonist VUAA1, as in the Hessian fly, Mayetiola destructor (Mdes) (Andersson et al., 2016; Corcoran et al., 2018), which may explain why the transfection of AcerOr1 can result in sensitivity to odorants in the absence of AcerOr2. However, when AcerOr2 is transfected into Sf9 cells with expression of only AcerOr2 subunits alone in the presence of endogenous Orco, it can form a functional homodimer and act as a non-specific cation channel sensitive to VUAA1, which may indicate that VUAA1 directly activates AcerOr2 subunits. VUAA1 was previously shown to activate Orco from a number of insects including Anopheles gambiae, Heliothis virescens, and D. melanogaster (Chen, 2014; Jones et al., 2011; Turner et al., 2014). These results provide further evidence supporting the hypothesis that the binding site of the Orco subunits is functional when Orco is part of the OR complex (Chen, 2014), OrX and Orco form a heteromeric complex that acts as an odorant-gated cation channel, or Orco forms an onspecific homomeric complex that acts as a spontaneously opening cation channel with ionic permeability mostly for Ca2+ (Jones et al., 2011). Ca2+signaling is used in various pathways necessary for appropriate recognition of odorant. This scenario might result from stimulation by odorants and transmission of odor signals.

Insect OR–Orco forms a non-selective ligand-gated ionotropic channel, although metabotropic signaling by various second messenger cascades may also occur (Deng et al., 2011; Getahun et al., 2013; Nakagawa & Vosshall, 2009; Sargsyan et al., 2011; Touhara, 2009); however, a recent study found that Agam/Orco alone or in combination with odorant ligand-binding ORs form ion channels that do not have cyclic nucleotide sensitivity (Jones et al., 2011). It should be noted that the relationship between insect ORs and the second-messenger signal transduction pathways and downstream effector enzymes mediated by G-protein remains hitherto unconfirmed. However, in this paper we did not directly study the potential involvement of the second-messenger signal transduction pathways. The interaction between ligand-binding ORs and Orco was previously investigated using bioluminescence resonance energy transfer (German et al., 2013; Neuhaus et al., 2005), pull-down assays (Tsitoura et al., 2010) and yeast-2-hybrid assays (Benton et al., 2006), showing that the third intracellular loop probably mediates the heteromeric interaction between ligand-binding ORs and Orco.

Several signals regulate the opening and closing of ion channel gates, for example, protein–protein interactions, ligand binding, membrane electrochemical gradients, and post-translational modifications, and these signals are thought to cause conformation changes in the ion channels to expand or contract the gate (daCosta & Baenziger, 2013; Zhou & McCammon, 2010). To date, three signals (odorant binding, calmodulin binding, and phosphorylation) have been proposed as regulators of the Orco/OrX ion channel. In this study, we aimed to provide evidence for homomeric and heteromeric interactions of the ligand-binding ORs AcerOr1 and AcerOr2. Other signaling pathways that regulate the ORs AcerOr1 and AcerOr2 need to be investigated in future studies.

Although AcerOr1 responded to nine different compounds and AcerOr2 responded to VUAA1 delivered at a high concentration, they responded to far fewer compounds at lower concentrations. Our results show that AcerOr1 and AcerOr2 have different dose responses, different odorant compounds, and EC50 values of 10−7 to 10−8 M. Overall, our results for the EC50 values were higher than those described for Epiphyas postvittana receptors such as EpOR1 and EpOR3 (Jordan et al., 2009) and lower than those for D. melanogaster Or22a (Pelz et al., 2006; Kiely et al., 2007).

In many species, olfactory transduction is triggered by odorant ligands, which interact with ORs coupled with heterotrimeric G-proteins. The surprising inverted topology of AcerOrs in addition to the heteromerization/homomerization with Orco in vitro also raised a question of whether AcerOrs really interact with G proteins during the honeybee olfactory transduction process in vivo. It has not yet been determined whether there are unknown intracellular G protein binding motifs present on AcerOr1 or AcerOr2, or whether heteromeric OR–Orco complexes signal without using G protein-dependent processes. Thus, it is unclear whether the increase in calcium signaling is a result of the ionotropic property of the proteins or whether these proteins activate G-protein-dependent signaling leading to calcium elevation, we will address this in a future study.

However, in the present study, the antennal EAG response to three of the ligands showed dose-independent activity, which indicated that the selected odorants may affect honeybee behavior in vivo.

Conclusion

In summary, we identified that AcerOr1 and AcerOr2 are functional odorant receptors in A. cerana cerana. AcerOr2 is essential for olfactory signaling, and AcerOr1 is a broadly tuned receptor. The ability of AcerOr1 to recognize the compounds identified here might represent an adaption of honeybees towards screening of floral scents when foraging.

Supplemental Information

Supplemental Information 1 β-actin raw data in Fig. 1.

The expression of β-actin were determined by western blotting in Sf9 cells, pIB/V5-His transfected cells, recombinant His-tagged AcerOr1 transfected cells, and recombinant His-tagged AcerOr2 transfected cells.

Click here for additional data file.

Supplemental Information 2 AcerOr2 raw data in Fig. 1.

The expression of AcerOr2 were determined by western blotting in Sf9 cells, pIB/V5-His transfected cells, recombinant His-tagged AcerOr1 transfected cells, and recombinant His-tagged AcerOr2 transfected cells.

Click here for additional data file.

Supplemental Information 3 AcerOr1 raw data in Fig. 1.

The expression of AcerOr1 were determined by western blotting in Sf9 cells, pIB/V5-His transfected cells, recombinant His-tagged AcerOr1 transfected cells, and recombinant His-tagged AcerOr2 transfected cells.

Click here for additional data file.

We would like to thank Chunxiang Zhang, who provided technical assistance in the laboratory, and Professor Xianchun Li, who helped review the draft manuscript. We would like to thank Editage [www.editage.cn] for English language editing.

Additional Information and Declarations

Competing Interests

Author Contributions

Data Availability

The authors declare that they have no competing interests.

Lina Guo conceived and designed the experiments, performed the experiments, analyzed the data, prepared figures and/or tables, authored or reviewed drafts of the paper, approved the final draft, conceived and designed the experiments, performed the experiments, analyzed the data, wrote the manuscript, prepared the figure and table.

Huiting Zhao authored or reviewed drafts of the paper, reviewed drafts of the paper.

Yusuo Jiang contributed reagents/materials/analysis tools, authored or reviewed drafts of the paper, contributed reagents/materials/analysis tools, reviewed drafts of the paper.

The following information was supplied regarding data availability:

The raw data are included in the Supplemental Files.

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
