# Peer review of "Expressional and functional interactions of two Apis cerana cerana olfactory receptors"

_PeerJ, doi:10.7717/peerj.5005_

## Round 0.1 · original submission · Major Revisions

Reviewer 1 has a series of comments that are largely straight-forward to address. These should pose no significant issue.

I would request however that you pay attention in particular to comments of Reviewer 2 and Reviewer 3.

Both have questioned aspects of this study and you need to carefully consider your response. The issues surrounding whether the receptors are heteromers or homo-dimers is important and must be discussed and/or addressed for the data to be meaningful.

It is important that you give careful consideration to the requirement for figures, and the clarity of presentation. This is particularly true for figures 1, 3 and 4, as reviewer 2 is of the opinion that these are largely superficial analyses. I agree with the point on Figure 1, and think that Figure 3 and 4 could be merged or included as supplemental data. However both have expressed specific concerns regarding each figure and these must be carefully and rigorously answered.

The three reviewers have all commented on the length and clarity of the paper; you need to carefully consider each stylistic point and address them, particularly in regards to the length of each section, what should be included, and where. We strongly urge you to take advice on these points from an experienced English speaker. This is important as there is a concern that the message of your manuscript is being lost as a result.

Reviewer 1 ·

Basic reporting

The manuscript by Lina Guo et al entitled ‘Expressional and functional interactions of two Apis cerana cerana olfactory receptors’ describes the expression and pharmacological characterisation of odorant receptors found in the Eastern honeybee. The authors expressed the receptors in insect cell line and then performed a series of molecular, biochemical and signalling assays to determine their functions in an in vitro system and in the context of food sensing/antennae function. The olfactory receptors are considered as understudied group of GPCRs and the authors should be commended for performing these studies. I have some suggestions, which I believe will improve the manuscript.

In the introduction, the authors should include what is known about the signalling/G protein coupling preference of these odorant receptors.

It would be helpful if the authors show the amino acid sequences of the two receptors and highlight the regions which were targeted for the generation of the antibody and explain why these regions were chosen.

The manuscript has series of mistakes, listed below which should be corrected:

Grammatical errors and missing words in sentences such as seven receptors which should read seven transmembrane receptors, conclution which should read conclusion etc.. Please carefully read the manuscript and make corrections.

Figure 1: Panel A, texts appear squashed and receptor band (second from right) for OR2 is misaligned - please rectify. Panel B, please include molecular weight in the blot.
Figure 2: Panel C, some cells appear yellow. Why? Please rectify.
Figure 3: Labelling (e.g. -, +) appear misaligned - please correct. Also please confirm in figure legend if the data was calculated relative to a house keeping gene (eg.beta actin) using the deltadelta to the power -2 analysis as indicated in the method.
Figure 4: Panel B and C please include the labelling in the quantification graphs in the same way as panel A. Also, include molecular weight for the blots and the antibody used in the blots (either in the figure itself or figure legend).
Figure 6. Please make the tables bigger and also include the maximum responses (efficacy) of the ligands. Please provide a brief explanation in the figure legend how the data was analysed. It says % maximal response. Which ligand and at what concentration the authors use as the reference/maximal response here?
Figure 8. It is not obvious in the text or the figure legend what the labelling on top of each bar graph (a,b,cd etc..) means. Please provide explanation in either the results section or figure legend.
Table 1: Also include the maximal responses (efficacy) in addition to potency (EC50).

Experimental design

Whilst the experiments were well designed, there are information which the authors should include/mention:

The authors should provide a brief rationale for measuring calcium output. Is calcium signalling considered to be the primary signalling pathway relevant for antennae function/sensing or is calcium assay used simply to narrow down the ligand choices?

The authors should also include a non transfected control in this assay.

The authors should provide the source of the receptor construct. Did the authors clone them from the honeybee or are they synthetically made/purchased from a company?

The authors should provide information on the molecular weight of AcerOr1 and AcerOr2 on the western blot data. Does the molecular weight correlate with the nominal molecular weight predicted from the DNA sequence? Or is it altered as to indicate posttranslational modification?

Validity of the findings

I have questions regarding the following observation:

Activation of AcerOr1 with a range of odorant molecules resulted in calcium elevation, a response generally thought to be through Gαq G-proteins. This response is augmented in the presence of the co-receptor AcerOr2 for a number of compounds tested.

1. Can the authors provide evidence that the two receptors interact via direct/indirect mechanism by performing co-IP etc..

2. Is the augmented response maintained in other signalling pathways (such as ERK, IP3, cAMP, arrestin recruitment, receptor phosphorylation/internalisation) linked to AcerOr1.

Reviewer 2 ·

Basic reporting

The text needs editing and typo corrections. This needs a professional input into the presentation style and content.

The discussion of the heteromeric receptors as being distinct has not been distinguished from differences in receptor expression levels and solely homomeric functional receptors.

A more focused article is needed.

Experimental design

Seems fine, but the evidence of expression is poorly presented and the justification for the knockout experiments is unclear (it reads like it is just extra data to pad the article). The evidence for heteromeric receptor function needs to be more rigorous.

Validity of the findings

Findings not validated in respect to heteromeric receptor expression. Most of the findings are not important, so the authors need to focus on what is actually worth saying and only provide that data required to substantiate that evidence.

Additional comments

Main criticism: The only data providing any new knowledge is the figures 5, 6 and 8. The rest is very pedestrian and superficial. Moreover, the data has not been used to demonstrate evidence of heteromeric expression.

L19: Transient transfection and cell surface expression of AcerOr1/2 in sf9 cells is useful, but why is it interesting to then knock out their expression with dsRNA?

L60: The argument for requiring sf9 cells is not robust. Please expand the argument by pointing out specific advantages. Many neuronal receptors from have been studied extensively in unrelated cell lines (HEK293, HeLa, COS-7) and these have yielded most of what we know about their pharmacology.

L65: The paragraph on RNAi is superficial and not relevant.

L70: The paragraph is not related to actions studied in this paper. Better would be a discussion of what is known about which receptors respond to which ligands.

L99: Vector construction does not provide sufficient detail for this to be repeated.

The methods read more like lab protocols that published methods. What is the %PFA used? This is important if the PM integrity needs to be intact to detect surface only staining (PFA may permeabilise cells, depending on concentration and duration.

Figure 1. DNA digest panel is trivial and should be removed. Showing only the selected band is undesirable and the whole lane needs to be shown for clarity (perhaps in the supplemental figures), otherwise specificity is unclear. Are the receptors devoid of N-glycosylation?

Figure 2. A higher magnification is required to demonstrate cell surface staining.

Figure 3/4 is useful if the knockout approach were to be used in native cells, to demonstrate that it works. It is not interesting when done on a cell line alone as this is expected.

Figure 5 is interesting but I am unsure as to what the co-expression results mean. Does this suggest a heteromeric receptor? Or is this equivalent to two separate homomers? Interpretation of this requires the relative expression levels between the cell lines to be known. The text claims that this figure indicates expression level, but it does not as this implies that function is directly proportional to expression, regardless of composition of Or.

Figure 6/7. The same applies as referred to in figure 5. Interesting but what does the co-expression data mean (hetero or 2 homos)?

Figure 8. No indication of error for n=3. A dose response curve (with error bars) should be given instead of panel A, which is then just repeated in panels B-D. If plotted on the same graph the relative potency will be clearer.

Table 1. Does the significance justify the exact values give for EC50, especially when the error does not match this confidence)? I doubt that very much (eg. 4.975 +/- 0.45.

Discussion. This section is way too long for the data presented, mostly because it repeats the results. The discussion claims evidence of heteromeric expression but this has not been demonstrated to me.

Q. Are sf9 cells devoid of endogenous Or expression? So that homomeric receptors are confirmed.

It strikes me that the authors are inexperienced at submitting professional research articles and would benefit from local expert advice.

·

Basic reporting

The introduction provides the reader with a good scope of the literature and expression systems other researchers have used to functionally explore insect ORs however there are areas that require further clarification and re-writing –

The opening paragraph of the introduction (line 33-40) in general is confusing. Line 38/39 –the authors state ‘These seven receptors are not related to any other receptor family’ – I think what the authors are referring to here are the fact that GPCRs are seven transmembrane spanning receptors. This becomes more confusing when the author goes from ‘seven receptors’ here to ‘hundreds of ORs’ from line 41 onwards. The authors need to be clearer as to what they mean. This needs to be re-written for clarity.

Line 76, could the authors elaborate here and give examples of the tissues that have been found to possess different OR levels.

The author covers AmOr11 function in the honey bee (line 70 onwards) however from here it is not quite clear why AcerOr1 and AcerOr2 was chosen for investigation – I think this needs to be explained and covered better in the introduction (maybe ideally placed between line 78 and 79).

Experimental design

The methodology section provides a good level of detail and experimental information for repetition of conditions.

Major – Can the authors insert a section to inform the reader of how they processed Western blotting and conducted densitometry and any normalisation carried out (if any) against β actin loading controls.

Minor corrections/updates –
Line 120 – could the authors state the concentration of G418 required to generate the stable cell lines.
Line 132 – could the authors state the concentration of PFA used to fix their cells.
Line 144 – could the authors state the components of the lysis buffer used for Western blotting experiments.

Validity of the findings

Results –
Figure 1 – title is generic and not very informative. ‘Effects of transfection’ is vague. Could the authors reword this so that the specific effects being investigated is more clearly stated. The images in figure 1A and B appear to be representative. Could the authors please indicate in the legend how many repetitions of these experiments were conducted. Figure 1A – the font size on the lane labelling needs to be increased so it can be read more clearly. Figure 1B – The alignment of the wells presented for each of the panels do not match up to one another. Could the authors state if these images are reprobes from the same membranes or sample re-runs. There is no control panel (example β actin) to demonstrate equal loading across the sample wells. This needs to be included. OR protein size information needs to be included either in the figure or legend information.

Figure 2 – Could the authors please indicate in the legend how many repetitions of these experiments were conducted. The subcellular localisation of AcerOr1 and Acer Or2 in the IF images provided is not entirely clear at the current magnification. Could the authors provide images that offer increased magnification so that subcellular features of the receptor are more clearly depicted.

Figure 3 – The annotation in the graph presented in Fig 3A needs to be better aligned to the corresponding bars. The authors have *** in the figure panels but the figure legend only indicate P* and P**. Can the authors define the P*** in the legend to coincide with their presence on the graphs and the statistical analysis used with any post test conducted.

Can the authors comment upon the low expression levels of AcerOr1 mRNA and AcerOr2mRNA in Fig 3E/F (without dsRNA) compared to the other panels (A-D). Also, the dsAcerOr1 seems to be more effective and selective for knockdown of the AcerOr2 (fig 3D) compared to Fig 3B, yet co-treatment with dsAcerOr1/dsAcerOr2 is most variable (3E/F).

Figure 4 – Can the authors annotate the blots in A, C and D to indicate protein band size for each panel. The β-actin blot in Fig 4A looks upside down. Based upon the representative AcerOr2 panel in Fig 4B, I am not convinced that AcerOr2 is significantly reduced. The quality of the blots in the AcerOr2 panel is poor and the βactin panel lane 3 appears to be more reduced than the AcerOr2 levels which raises doubt as to the equal loading of protein between samples. Can the authors define the P*** in the legend to coincide with their presence on the graphs and the statistical analysis used with any post test conducted for these experiments.

For the combination dsRNA experiments, the authors have chosen 500ng of dsRNA yet these conditions did not appear to significantly affect mRNA levels in the experiments in Fig 3 F. Could the author comment upon this please.

I would advise the authors to revise lines 278-286. The data presented in Figure 5 A/B/C needs to be described better in the results text. There is insufficient information to support the figure as to the experiments conducted and the nature of compounds selected for screening. Based on the results text it is not clear what cells are being tested in panels A-C. This needs to be signposted better. The title in Figure 5 needs corrected – remove ‘a’ in ‘to a various’. The information in the legend states that bars indicate standard deviation then they state that data points represent mean/SEM. Can the authors clarify what they define as bars versus data points. This is confusing and needs to be clearer.

Given that the authors provide a dedicated table for all EC50 values (table 1), I would ask the authors to remove the individual tables from beneath panel A-C. The authors need to make sure that they insert the units into the EC50 values in the results text where appropriate (line 287 onwards). The units should also be clearly presented in Table 1. The curve fitting software used to derive the EC50 values obtained in Figure 6A-C needs to be inserted into the figure legend and the nature of any equations used (i.e. Hill equation). The information in the legend related to the panels need to be corrected. Based on the legend information, the authors have mixed up Figure panel B and C. The authors need to correct the legend information – there are no bars presented here yet the authors state ‘Bars indicate standard deviation based on three independent experiments’. Please edit accordingly to accurately reflect the data presented.

The results text related to Fig 6 (line 287-292) is very brief and does not support the figure content very well. The EC50 values cited for linolenic acid (line 289) in the results text are different to the values presented in Table 1. It is not clear why the authors have focussed specifically on that one EC50 value.

Figure 7 needs to be described better in the results text. The authors need to provide information related to the concentration of odorants used to perform the experiments conducted in Figure 7.

Major comment – The excitation and emission wavelengths for Fluo4 and GFP are similar which often leads to interference in calcium flux assays when cells express GFP. The authors used dsGFP as a control in their experiments, can the authors comment upon how the presence of GFP in their cell system impacted their calcium imaging. This needs to be clearly stated.

Figure 8 – Line 305: Based upon the data obtained so far, it is not clear why these three odorants were selected for further investigation. This needs to be explained better in the text. The results text that corresponds to figure 8 is minimal and insufficient to support the data presented. The same applies to the information provided in the figure legend. In its current form, the legend does not support the data presented in panels A-D. Despite the information provided related to the different letters at the top of each bar, it is not clear what a, b, c, d, cd actually correspond to by way of ‘significance’. The authors need to revise and re-write the content of the results text and legend to make sure that this is clearer.

Minor - Typographical error, Line 412 - Conclusions

Additional comments

Whilst the authors cover a vast body of literature throughout the manuscript related to the importance of ORs in odor recognition processes for a host of insects, one of the aspects that is lacking in detail are the known signal transduction pathways triggered upon OR activation in response to odorants. Information related to the G-proteins that couple to ORs and their second messenger pathways activated really needs to be included. The conclusions from the study related to odor recognition for AcerOr1 and AcerOr2 relies upon data derived principally from calcium imaging experiments. This is really only one output that has been explored therefore the ability of these odorants to trigger other signalling pathways downstream of AcerOr1 and AcerOr2 is still unknown. The authors need to state why this pathway was considered to be the most appropriate for the receptors studied. In the GPCR field, ligand bias and bias signalling is becoming more apparent. Whilst the ligands tested for the AcerOr2 expression systems only revealed VUAA1-mediated calcium activity, the potential for other pathways to be activated downstream of AcerOr2 (other than calcium) by the other odorants has to be considered and discussed. This is a major limitation of the current investigation as no other signalling pathway was explored.

---

## Round 0.2 · Minor Revisions

Thank you for submitting your revised paper which largely addresses all the points raised by the previous reviewers. As these were substantial, I returned the paper to them for comment. Both have made a few additional comments which are enclosed. The revisions requested are largely trivial, and should not take you very long. However, I would urge you to pay particular attention to aspects of the English which in places requires attention.

Reviewer 1 ·

Basic reporting

The manuscript can be improved by making the sentences easier to follow and correcting sentences that do not make sense such as "So functional characterization of AcerOr1 do not through Gαq G-proteins can also stably transformed Sf9 cells". There are also some confusing sentences especially whether Or1 and Or2 are behaving as ion channels or G-protein coupled receptors. As such, it is unclear whether the increase in calcium signal is a result of the ionotropic property of the proteins or that these proteins activate G-protein dependent signalling leading to calcium elevation. The authors should clarify this.

Experimental design

Experimental design seems fine. There are minor mistakes that can be addressed easily such as the primers for AcerOr2 which are written as both 'forward primer' and the calcium response which does not have a unit.

Validity of the findings

The authors need to be cautious with regard to dimers (e.g. Or2+Or2 leading to the formation of ion channel) especially that no evidence is provided in the manuscript.
The observation that calcium signalling is augmented when Or1 is co-expressed with Or2 is interesting and may play a role in fine tuning the sensitivity of odor sensing. However the authors do not provide evidence how this is achieved. The authors should provide evidence for the idea that these proteins form dimers by performing additional experiments such as designing minigene to disrupt the dimer or co-immunoprecipitation. Otherwise the co-transfection data becomes uninterpretable and should be omitted. Any mentions of a dimer should also be removed.

Additional comments

Please carefully revise the manuscript to make it easier to follow.

·

Basic reporting

The manuscript has been dramatically changed since it's original submission in a way that has improved the clarity of the research findings. I feel that the authors have made a serious attempt to address most of the concerns raised from the previous peer reviewers.

I have minor comments that I feel could be easily addressed by the authors -
The abstract needs to be improved from Line 19-23. The author needs to be more clear on what exactly revealed 'distinct changes to concentration response curves' and what those distinct changes were.
Can the authors provide details as to the signalling cascade that results in opening of ion channels (line 30-31).
Remove 'is' on line 54.
Amend the font of the text in line 171
The reference on line 178 is incomplete and requires a date.
The results text from line 246-256 needs to be organised better. Whilst it is great to have the numerical information, there has to be a better balance of descriptive text rather than just relay EC50 values compiled in table 1.
Lines 290-291 need to be written better - this line in its current form it does not make sense.

Figure 1 title - check the text for formatting - It is not clear what the title of this figure is.
For some reason the Western blot panels look over-processed (A) and the images in panel B look as if the signals are extremely high.
Figure4 legend for (A) mentions 'at the same concentration of 10-10M' ... this is confusing given the data presented in panel A which is a dose-response curve. This needs to be corrected.
The formatting of Table 1 needs more careful consideration so that the information is presented clearly.

Experimental design

The research question has been simplified in a way that unnecessary data has been removed. I feel this has made the paper more straightforward.
The methods has been updated to include the key information for repetition.

Validity of the findings

I feel the authors need to elaborate further on the point they made in their discussion regarding what their data indicate. On lines 356-359 the authors state that their results indicate that AceOrs could function as a G protein-coupled receptors in vitro. The functional data that is presented in the paper is merely demonstrating calcium response EC50 values so I feel the authors need a stronger argument for this statement to be true. At no point do the authors really explore the GPCR nature of the AcerOrs. This point needs to be better addressed at the experimental level before such claims can be made. I would urge the authors to consider this point in their discussion.

---

## Round 0.3 · accepted · Accept

The changes made to your revised manuscript have addressed all the minor points raised and hence the recommendation is to accept the paper.

#